# Patterns of health service utilisation of mothers experiencing mental health problems and intimate partner violence: Ten-year follow-up of an Australian prospective mother and child cohort

Deirdre Gartland[1,2], Kelsey Hegarty[3,4], Sandra Papadopoullos[1], Stephanie Brown[1,2,3]*

1 Intergenerational Health, Murdoch Children's Research Institute, Melbourne, Australia, 2 Department of Paediatrics, The University of Melbourne, Melbourne, Australia, 3 Department of General Practice, The University of Melbourne, Melbourne, Australia, 4 Royal Women's Hospital, Melbourne, Australia

* stephanie.brown@mcri.edu.au

**Data Availability Statement:** The conditions of ethics approval preclude data access via a public repository. Data sharing is subject to approval by

## Abstract

### Objectives

Few studies have investigated health service use of mothers experiencing mental health problems or intimate partner violence (IPV). The aim of this study was to investigate health service utilisation of mothers experiencing mental health problems and intimate partner violence ten years after having a first baby.

### Methods

Prospective cohort of 1507 first-time mothers recruited in Melbourne, Australia. Follow-up at ten years incorporated: Center for Epidemiologic Studies Depression Scale, Beck Anxiety Inventory, Posttraumatic Stress Disorder Checklist, Composite Abuse Scale.

### Results

At ten years postpartum, one in four mothers (26.1%) reported depressive, anxiety or post-traumatic stress symptoms, and almost one in five (19.4%) reported recent IPV. Two-fifths of mothers reporting clinically significant mental health symptoms had experienced recent IPV (Odds Ratio = 5.6, 95% CI 3.9–8.1). Less than half of mothers experiencing mental health problems at ten-year follow-up had discussed their mental health with a general practitioner and around one in three had talked to a mental health professional. Two-thirds of mothers experiencing recent IPV had not disclosed this to a general practitioner or mental health professional.

### Conclusions

The findings highlight the extent to which many women deal with IPV and mental health problems without the support that primary health care and mental health care could provide

the investigator team. Applications will be considered in context of papers in progress, and compliance with conditions of ethics approval and consent. Interested researchers are invited to submit a request via the Melbourne Children's LifeCourse platform (https://lifecourse. melbournechildrens.com/#lifecourse-data-and-access) or to contact the Principal Investigator, Professor Stephanie Brown (stephanie. brown@mcri.edu.au).

**Funding:** The Maternal Health Study has been supported by three project grants from the Australian National Health and Medical Research Council (NHMRC, 191222 awarded to SB; 433006 awarded to SB and KG, and 1048829 awarded to SB, DG and KH); grants from the William Buckland Foundation, ANZ Trustees Medical Research Fund and Australian Rotary Health, awarded to SB; the Safer Families Centre of Research Excellence (NHMRC, 1116690) awarded to KH and SB and the Stronger Futures Centre of Research Excellence (NHMRC 1198270) awarded to SB and DG. Research conducted at the Murdoch Children's Research Institute is supported by the Victorian Government Operational Infrastructure Program. There was no additional external funding received for this study. The funders had no role in study design, data collection and analysis, decision to publish, or preparation of the manuscript.

**Competing interests:** The authors declare no competing interests.

and point to the need for more concerted efforts to strengthen health system responses to these frequently related issues.

## Introduction

Poor maternal mental health and intimate partner violence (IPV) are major public health issues linked to intergenerational cycles of trauma and social adversity [1, 2]. In high income countries, 10–20% of mothers experience depressive symptoms [3, 4] and 15% of mothers experience anxiety symptoms in the year after childbirth [5]. Symptom prevalence remains high throughout the first ten years of motherhood, with several studies suggesting that maternal mental health problems increase in middle childhood [6–8]. Intimate partner violence is estimated to affect one in three women at some stage in their lives [9], and more than one in three mothers in the decade after having their first child [6].

The World Health Organization has called for systems change to strengthen health sector responses to IPV and poor maternal mental health [10]. In Australia, two recent Royal Commissions have drawn attention to the need for systems reform to improve prevention and early intervention to support women, children and families affected by family violence and mental health issues [11, 12]. Recent systematic reviews provide evidence of the multiple personal, organisational and systems barriers to disclosure of IPV and mental health problems [13–15]. Organisational and systems barriers include: low affordability of services; lack of availability of transport and/or childcare; poorly developed systems for identification of women experiencing mental health problems and/or IPV; fragmented referral pathways; inadequate provision of language services; insufficient attention to tailoring of care for culturally diverse communities; and lack of integration of health and social care [14, 16]. Personal barriers include: minimising the significance of the problem; the belief that nothing will help; shame, self-blame and low self-esteem; and concerns about the consequences of disclosure, including escalation of violence and the risk of child removal [14, 17].

While many qualitative studies have explored organisational, systems and personal barriers that limit women's access to health care in relation to both maternal mental health symptoms and IPV, few population-based longitudinal studies document patterns of health service utilization comparing mothers experiencing IPV with those not experiencing IPV or examining the experiences of women with mental health problems compared with those who do not. Daoud and colleagues compared health care utilization patterns in a cohort of perinatal women using record linkage to examine health care utilization of Jewish and Arab minority women in Israel. Experiencing IPV was associated with higher health care utilization among Arab women and lower utilization among Jewish women leading the authors to conclude that tailored solutions are needed to meet the needs of ethnically diverse women [18, 19]. This study was not able to identify the extent to which women experiencing IPV disclosed this to health professionals. Drawing on data from a prospective pregnancy cohort, Woolhouse and colleagues reported that mothers were more likely to seek support from health professionals regarding depression than anxiety in the first nine months postpartum, and more likely to discuss and seek support in relation to mental health than they were to discuss or seek support in relation to experiences of IPV at four years postpartum [18–20].

The current study is the first to (i) examine patterns of health service utilization of mothers experiencing depressive, anxiety and posttraumatic stress symptoms and IPV ten years after the birth of their first child, and (ii) report data on the extent to which mothers experiencing

clinically significant mental health symptoms and/or IPV discussed their mental health or IPV with health professionals during consultations. The paper draws on data collected in an Australian prospective pregnancy cohort of first-time mothers followed up ten years after their first birth. The major aim of the paper is to investigate the extent to which mothers experiencing mental health symptoms and recent IPV seek assistance from general practitioners, mental health professionals and advocacy services (e.g. legal services). We hypothesised that women would be more likely to talk to health professionals about their mental health symptoms than they would be to discuss IPV, as Australian primary care services are better equipped to recognise and respond to maternal mental health than they are to identify and support women experiencing IPV [11]. We also hypothesised that women experiencing physical IPV would be more likely to receive support from health professionals than women experiencing emotional IPV alone. This hypothesis was based on studies demonstrating that women experiencing more severe forms of IPV are more likely to tell someone about it [17] and evidence of greater community recognition of physical IPV [21].

## Materials and methods

### Participants

First-time mothers were recruited via six public hospitals in Melbourne, Australia between April 2003 and December 2005. Women were eligible to take part if they were: nulliparous, ≤ 24 weeks' gestation, aged ≥18 and sufficiently proficient in English to complete study questionnaires. Women were mailed an invitation by maternity hospitals and were also provided with information about the study during antenatal appointments and childbirth education classes. Follow-up included questionnaires at one, four and ten years postpartum. In this paper, data are drawn from questionnaires completed in early pregnancy and at ten years postpartum. Ethics approval was obtained via the research ethics committees of participating hospitals, La Trobe University and the Royal Children's Hospital, Melbourne, Australia. Written consent was obtained from all women participating in the study.

### Study measures

**Maternal mental health.** At ten years postpartum, measures of mental health included: the Center for Epidemiologic Studies Depression Scale (CES-D), the Beck Anxiety Inventory (BAI) and the Posttraumatic Stress Disorder Checklist—Civilian Version (PCL-C) [22–24]. The CES-D is a 20-item self-report measure of depressive symptoms. We used the cut-point of ≥20 which has been found to have good sensitivity and specificity for identifying probable major depressive disorder in community samples [25]. The BAI is a 21-item measure that asks about common symptoms of anxiety, such as feeling nervous or unable to relax. We used the recommended cut-off of ≥16 to identify clinically significant anxiety symptoms [23, 26]. The PTSD Checklist identifies symptoms of PTSD matching the DSM-IV criteria. We used the cut-point of ≥35 recommended for use in community samples to identify clinically significant symptoms [27]. Thoughts of self-harm were identified based on responses to item 10 of the Edinburgh Postnatal Depression Scale [28].

**Intimate partner violence.** Exposure to IPV in the 12 months prior to the 10-year follow-up was assessed using the 30-item Composite Abuse Scale (CAS) [29, 30]. The CAS is a comprehensive multidimensional measure of IPV that includes items asking about emotional abuse (e.g. 'Blamed me for their violent behaviour') and physical abuse (e.g. 'Pushed, grabbed or shoved me') by a current or former partner. Women were asked to indicate how often these actions happened in the previous 12 months: *never, once, several times, once per month, once per week, or daily* (scored 0 to 5). Physical IPV was defined as a score of ≥1 for physical abuse

items and emotional IPV as a score of ≥3 for emotional abuse items. Women were defined as experiencing IPV if they scored positive for physical and/or emotional IPV.

**Health service use.** At 10-year follow-up, women were asked "How many times have you consulted a local doctor or GP (general practitioner) in the last 12 months for issues related to your own health?", with response options "never, once, twice, three times, four times, five to six times, seven or more times". In addition, women were asked "In the last 12 months, have you talked to a general practitioner, a mental health professional (e.g psychologist, psychiatrist, counsellor) or other health professionals" about: "tiredness", "feeling depressed or low", "stress or anxiety", "relationship problems" or "violence in your home" (yes/no). Finally, women were asked "Have you used any of the following services in the last 12 months: "legal services", "financial counselling", "housing services" or a "telephone helpline" (yes/no).

## Analysis

Data were analysed using STATA version 16 [31]. Representativeness of the original cohort was assessed by comparing data on obstetric and social characteristics of participants with routinely collected data for women giving birth as public patients in Victoria during the recruitment period. Analyses were initially conducted as complete case analyses drawing on data for participants who completed the 10-year follow-up, and then repeated using multiply imputed data to account for selective attrition over time. Forty data sets were imputed using chained equation modelling [32]. The imputation model included variables associated with attrition and variables significantly associated with mental health outcomes and/or exposure to recent IPV at 10-year follow-up. Patterns of association were similar in analyses with multiple imputation and complete case analyses, but higher prevalence estimates were obtained in imputed data sets. As imputed data are more likely to give a robust picture of prevalence of IPV and mental health symptoms at 10-year follow-up, results are presented using multiply imputed data (n = 1507) where possible.

Univariable logistic regression was used to examine social characteristics of women experiencing mental health symptoms and IPV. Data are presented as unadjusted Odds Ratios and 95% confidence intervals. Next, we estimated the prevalence and odds of recent IPV in those with and without mental health symptoms at 10-year follow-up. Data on number of visits with general practitioners in the previous 12 months (outcome variable) are presented as frequencies with 95% confidence intervals. In addition, we calculated adjusted odds ratios and 95% confidence intervals for: the number of visits made to general practitioners by women experiencing mental health symptoms, women experiencing IPV, and women experiencing mental health symptoms and IPV. The reference groups for these analyses were: women not reporting mental health symptoms, women not reporting recent IPV, and women not reporting mental health symptoms or recent IPV, respectively. Logistic regression models accounted for potential confounding by maternal age and possession of a health care card (as a marker of social disadvantage). We chose not to adjust for a broader range of covariates as analyses were conducted to establish patterns of health service use rather than to assess risk factors. Finally, data are presented on the frequency with which women experiencing mental health symptoms and/or IPV discussed specific health issues with general practitioners, mental health professionals and other health professionals, and the frequency of contacts with other services. These data are based on complete case analyses as the relatively small number of women discussing IPV with general practitioners (<5%) and similarly small number of women using financial counselling or telephone help lines meant it was not possible to estimate imputed values for these variables. All other data are reported based on data derived using multiple imputation.

## Results

A total of 1507 eligible women enrolled in the study. We estimate that one in three women invited to take part joined the study. It is not possible to calculate a precise response fraction as many women received invitations via more than one pathway. Comparison of participant characteristics with those of all nulliparous women giving birth at Victorian public hospitals during the study period showed that the sample was representative in relation to method of birth, infant birthweight and gestation, but that younger women (18–24 years of age, 14.1% versus 29.8%) and women born overseas of non-English speaking background (16.2% versus 21.5%) were under-represented [33, 34]. At enrolment (mean of 15 weeks' gestation), a majority of women were living with a partner and had completed some post-secondary education (e.g. certificate, diploma or university degree). Further information about sample characteristics is available in a recent study update [34]. Women who participated in 10-year follow-up (63.2% of the original cohort) were more likely to be older, Australian born, tertiary educated, not in receipt of a government health care card, and less likely to have reported IPV or depressive symptoms in the year following the birth of their first child (S1 Table).

### Characteristics of women experiencing mental health symptoms and IPV

The prevalence of depressive symptoms (CES-D $\geq$20) at 10-year follow-up was 18.7% (95% CI 16.0–21.4%). Anxiety symptoms (BAI $\geq$16) were reported by 12.5% (95% CI 10.0–14.9%) of women, posttraumatic stress symptoms (PCL-C $\geq$35) by 17.0% (95% CI 14.5–19.7%) and thoughts of self-harm by 8.4% (95% CI 6.4–10.5%) of women. Overall, more than one in four women (26.1%, 95% CI 23.1–29.0%) experienced depressive, anxiety and/or posttraumatic stress symptoms and almost one in five women (19.4%, 95% CI 16.6–22.3%) experienced IPV in the 12 months prior to 10-year follow-up. Emotional IPV was the most common type of IPV experienced by women in the study: 11.9% of women experienced emotional IPV alone, 6.4% experienced emotional and physical IPV and 1.1% experiences physical IPV alone.

At 10-year follow-up, women who were single, separated or divorced; in receipt of a government health care card; had a low family income and/or no post-secondary education were more likely to be experiencing mental health problems and IPV (Table 1). Women aged under 25 at the time of their first birth were also more likely to be experiencing mental health problems and IPV. In contrast, women with two or more children were less likely to be experiencing mental health problems and/or IPV. There were no differences in maternal mental health and/or exposure to IPV associated with maternal country of birth.

### Prevalence of IPV in women experiencing mental health symptoms

More than two-fifths of women reporting depressive, anxiety and/or posttraumatic stress symptoms had experienced IPV in the year prior to 10-year follow-up, representing more than a five-fold increase in odds of experiencing IPV compared to women scoring below clinical cut-off scores for mental health symptoms (see Table 2). Half of the women reporting thoughts of self-harm had experienced recent IPV (50.9%).

### Health service use of women experiencing mental health symptoms and/or IPV

The majority of women made at least one visit to a general practitioner (92.1%, 95% CI 90.2–93.7%) for issues related to their own health in the 12 months prior to 10-year follow-up (Table 3). Over half made three or more visits (53.8%, 95% CI 57.1–50.3%) and almost one in four saw a general practitioner on five or more occasions (23.5%, 95% CI 20.2–26.2%). Overall,

**Table 1. Characteristics of women experiencing mental health symptoms and recent IPV at 10-year follow-up (n = 1507).**

| | Cohort | Depressive, anxiety or posttraumatic stress symptoms | | | Recent IPV | | |
|---|---|---|---|---|---|---|---|
| | Col% [95%CI] | % | OR [95%CI] | p-value | % | OR [95%CI] | p-value |
| Maternal age* | | | | | | | |
| ≥ 25 at time of first birth | 56.9 [54.4–59.4] | 23.1 | 1.0 [ref] | | 16.7 | 1.0 [ref] | |
| < 25 at time of first birth | 43.1 [40.6–45.6] | 30.0 | 1.4 [1.1–1.9] | 0.015 | 21.2 | 1.3 [1.0–1.8] | 0.070 |
| Maternal country of birth* | | | | | | | |
| Australia | 74.4 [72.2–76.6] | 26.8 | 1.0 [ref] | | 19.5 | 1.0 [ref] | |
| Overseas | 25.6 [23.4–27.8] | 24.0 | 0.9 [0.6–1.2] | 0.392 | 16.3 | 0.8 [0.5–1.3] | 0.372 |
| Maternal highest educational qualification* | | | | | | | |
| Tertiary | 72.1 [69.8–74.4] | 24.2 | 1.0 [ref] | | 16.5 | 1.0 [ref] | |
| Year12 or less | 27.9 [25.6–30.2] | 31.0 | 1.4 [1.0–2.0] | 0.044 | 24.2 | 1.6 [1.1–2.3] | 0.009 |
| Relationship status** | | | | | | | |
| Living with partner | 81.8 [79.3–84.4] | 21.6 | 1.0 [ref] | | 13.7 | 1.0 [ref] | |
| Single/Separated/Divorced | 18.2 [15.6–20.7] | 46.5 | 3.2 [2.2–4.6] | <0.001 | 41.3 | 4.4 [3.0–6.7] | <0.001 |
| Total family income per annum (AUD)** | | | | | | | |
| >100,000 | 42.1 [39.1–45.2] | 19.1 | 1.0 [ref] | | 9.4 | 1.0 [ref] | |
| >60,001–100,000 | 30.3 [27.4–33.2] | 23.1 | 1.3 [0.8–2.0] | 0.274 | 17.5 | 2.0 [1.3–3.3] | 0.004 |
| <= 60,000 | 27.5 [24.6–30.4] | 40.2 | 2.9 [1.9–4.2] | <0.001 | 34.2 | 5.0 [3.2–7.8] | <0.001 |
| Health care card ** | | | | | | | |
| No | 76.7 [73.8–79.6] | 21.3 | 1.0 [ref] | | 13.5 | 1.0 [ref] | |
| Yes | 23.3 [20.4–26.2] | 41.9 | 2.7 [1.9–3.8] | <0.001 | 35.9 | 3.6 [2.4–5.3] | <0.001 |
| Parity** | | | | | | | |
| One child | 19.3 [16.4–22.1] | 38.0 | 1.0 [ref] | | 28.9 | 1.0 [ref] | |
| Two or more | 80.7 [77.9–83.6] | 23.3 | 0.5 [0.3–0.7] | <0.001 | 16.2 | 0.5 [0.3–0.7] | <0.001 |
| Total | 100 | 26.1 | | | 19.4 | | |

* Maternal age, country of birth and educational qualifications during pregnancy/early postpartum.

** Relationship status, family income, health care card status and parity were assessed at 10-year follow-up.

women with mental health problems made more visits to general practitioners than women without mental health problems. More than two-thirds of women with depressive symptoms or posttraumatic stress symptoms and three quarters of women with anxiety symptoms made three or more visits to a general practitioner. Two-fifths of women with depressive or posttraumatic stress symptoms and almost half of women with anxiety symptoms made five or more visits to a general practitioner, compared to around half of women without mental health symptoms. Using two or less visits as the reference category, there was a two to four-fold increase in odds of women experiencing mental health problems making five or more visits to a general practitioner compared with women not experiencing mental health problems. Women experiencing recent IPV also made more visits to a general practitioner than women not experiencing IPV, although this trend was not as pronounced as for women experiencing mental health symptoms. One in three women (36.6%) experiencing recent IPV made two or less visits to a general practitioner, and over half (63.4%) made three or more visits. One in three women experiencing recent IPV (36.3%) made five or more visits to a general practitioner equating to almost a two-fold increase in odds of making five or more visits compared to women not experiencing IPV. Two-fifths of women experiencing physical IPV made five or more visits to a general practitioner (39.5%, 95% CI 24.5–54.6%), compared with one third of women (34.2%, 95% CI 23.5–44.9%) experiencing emotional IPV alone.

**Table 2. Likelihood of experiencing recent IPV among those with and without mental health symptoms at ten years postpartum (n = 1507).**

|  | Cohort | Recent (past year) IPV | | |
|---|---|---|---|---|
|  | % [95% CI] | % | OR [95%CI] | p-value |
| Depressive symptoms (CES-D ≥20) |  |  |  |  |
| No | 81.6 [78.9–84.3] | 13.1 | 1.0 [ref] |  |
| Yes | 18.4 [15.7–21.1] | 43.3 | 5.1 [3.3–7.9] | <0.001 |
| Anxiety symptoms (BAI ≥16) |  |  |  |  |
| No | 87.5 [85.1–89.9] | 14.4 | 1.0 [ref] |  |
| Yes | 12.5 [10.1–14.9] | 48.1 | 5.5 [3.5–8.6] | <0.001 |
| Posttraumatic stress symptoms (PCL-C ≥35) |  |  |  |  |
| No | 83.5 [80.7–86.3] | 12.7 | 1.0 [ref] |  |
| Yes | 16.5 [13.7–19.3] | 48.7 | 6.6 [4.4–9.8] | <0.001 |
| Thoughts of self-harm (EPDS Item 10) |  |  |  |  |
| No | 91.7 [89.7–93.7] | 15.7 | 1.0 [ref] |  |
| Yes | 8.3 [6.3–10.3] | 50.9 | 5.6 [3.2–9.7] | <0.001 |
| *Depressive, anxiety and/or posttraumatic stress symptoms* |  |  |  |  |
| *No* | *74.4 [71.4–77.4]* | *11.0* | *1.0 [ref]* |  |
| *Yes* | *25.6 [22.6–28.6]* | *40.9* | *5.6 [3.9–8.1]* | *<0.001* |

**Table 3. Patterns in the number of visits to general practitioners in the 12 months prior to 10-year follow-up for women experiencing mental health symptoms and recent IPV (n = 1507).**

|  | Number of visits to GP | No | Yes | Adj. OR* [95%CI] | p-value |
|---|---|---|---|---|---|
|  |  | % | % |  |  |
| Depressive symptoms (CES-D≥20) | 0–2 visits | 50.2 | 29.2 | 1.0 [ref] |  |
|  | 3–4 visits | 30.1 | 31.9 | 1.7 [1.1–2.5] | 0.015 |
|  | 5+ visits | 19.6 | 38.9 | 2.9 [1.8–4.4] | <0.001 |
| Anxiety symptoms (BAI ≥16) | 0–2 visits | 49.9 | 21.3 | 1.0 [ref] |  |
|  | 3–4 visits | 30.3 | 31.4 | 2.2 [1.2–4.1] | 0.010 |
|  | 5+ visits | 19.8 | 47.3 | 4.8 [2.7–8.5] | <0.001 |
| Posttraumatic stress symptoms (PCL-C ≥35) | 0–2 visits | 50.5 | 26.0 | 1.0 [ref] |  |
|  | 3–4 visits | 30.3 | 31.5 | 1.9 [1.2–3.0] | 0.010 |
|  | 5+ visits | 19.3 | 42.4 | 3.7 [2.2–6.2] | <0.001 |
| *Depressive, anxiety and/or posttraumatic stress symptoms* | 0–2 visits | 52.2 | 29.6 | 1.0 [ref] |  |
|  | 3–4 visits | 29.8 | 32.4 | 1.8 [1.2–2.6] | 0.003 |
|  | 5+ visits | 18.0 | 37.9 | 3.1 [2.1–4.6] | <0.001 |
| Recent intimate partner violence | 0–2 visits | 48.6 | 36.6 | 1.0 [ref] |  |
|  | 3–4 visits | 31.3 | 27.1 | 1.0 [0.6–1.5] | 0.943 |
|  | 5+ visits | 20.1 | 36.3 | 1.8 [1.1–2.9] | 0.018 |
| *Mental health symptoms and recent intimate partner violence* | 0–2 visits | 48.4 | 29.8 | 1.0 [ref] |  |
|  | 3–4 visits | 30.8 | 27.7 | 1.3 [0.7–2.2] | 0.405 |
|  | 5+ visits | 20.8 | 42.5 | 2.5 [1.4–4.4] | 0.001 |
|  |  | 100.0 | 100.0 |  |  |

* Adjusted odds shown in the table are for the odds of making 3–4 visits or 5 or more visits to a general practitioner compared with the reference category of 0–2 visits for each sub-group of women in the sample. Odds Ratios are adjusted for maternal age and social disadvantage (health care card).

Table 4 (based on complete case analyses) shows the proportion of women experiencing mental health problems and/or IPV who had talked to health professionals about their mental health symptoms, relationship difficulties or IPV in the previous 12 months. A higher proportion of women with mental health problems had discussed feeling depressed or low with a GP: 35.5% had talked to a GP compared with 25.1% who talked to a mental health care professional. Almost half of women with a mental health problem (48.8%) had discussed stress/anxiety and/or feeling depressed/low with a GP compared to 33.0% with a mental health professional. In contrast, a higher proportion of women experiencing IPV had talked to a mental health professional about relationship issues and/or violence at home, than had talked to a general practitioner (27.5% versus 16.7%). In further analyses of these data, women experiencing physical IPV appeared to be no more likely to discuss IPV or relationship difficulties with their general practitioner (16.3% versus 17.3%, OR– 0.9, 95% CI 0.4–2.5) or mental health professional (30.6% versus 27.2%, OR = 1.4, 95% CI 0.6–3.1), compared with women experiencing emotional IPV alone. Relatively few women—less than 10%—nominated other health professionals that they had spoken to about their mental health and/or partner violence.

Seventy-three percent of women experiencing IPV (103/138) had *not* talked to a mental health professional about relationship difficulties or partner violence, and 83.3% (115/138) had *not* talked to a general practitioner. Almost two thirds of women experiencing IPV had *not* talked to any health professional about relationship difficulties or partner violence (65.2%, 90/138). In contrast, just over a third of women experiencing mental health problems had *not* talked to a health professional about their mental health (38.4%, 78/203).

**Table 4. Issues discussed with health professionals in the year prior to 10-year follow-up (n = 952)**[*].

| | Issues discussed | General practitioner | | Mental health professional | | Other health professional | |
|---|---|---|---|---|---|---|---|
| | | n | % | n | % | n | % |
| **Women with depressive, anxiety or posttraumatic stress symptoms (n = 203)** | Tiredness/exhaustion | 87 | 42.9 | 42 | 20.7 | 16 | 7.9 |
| | Stress/anxiety | 90 | 44.3 | 64 | 31.5 | 12 | 5.9 |
| | Feeling depressed/low | 68 | 33.5 | 51 | 25.1 | 5 | 2.5 |
| | Relationship problems | 23 | 11.3 | 42 | 20.7 | 2 | 1.0 |
| | Violence in home | 5 | 2.5 | 11 | 5.4 | 1 | 0.5 |
| | *Did not talk about these issues* | 86 | 42.4 | 134 | 66.0 | 182 | 89.7 |
| **Women experiencing recent intimate partner violence (n = 138)** | Tiredness/exhaustion | 55 | 39.9 | 22 | 15.9 | 7 | 5.1 |
| | Stress/anxiety | 48 | 34.8 | 37 | 26.8 | 4 | 2.9 |
| | Feeling depressed/low | 36 | 26.1 | 31 | 22.5 | 6 | 4.3 |
| | Relationship problems | 21 | 15.2 | 35 | 25.4 | 4 | 2.9 |
| | Violence in home | 6 | 4.3 | 17 | 12.3 | 4 | 2.9 |
| | *Did not talk about these issues* | 69 | 50.0 | 95 | 68.8 | 126 | 91.3 |
| **Women experiencing mental health symptoms and intimate partner violence (n = 67)** | Tiredness/exhaustion | 29 | 43.3 | 15 | 22.4 | 4 | 6.0 |
| | Stress/anxiety | 27 | 40.3 | 20 | 29.9 | 3 | 4.5 |
| | Feeling depressed/low | 24 | 35.8 | 17 | 25.4 | 4 | 6.0 |
| | Relationship problems | 12 | 17.9 | 18 | 26.9 | 1 | 1.5 |
| | Violence in home | 4 | 6.0 | 10 | 14.9 | 1 | 1.5 |
| | *Did not talk about these issues* | 32 | 52.2 | 45 | 67.2 | 62 | 92.5 |

[*] Table reports complete case analyses. Totals do not add to 100% as women could talk about more than one health issue with health professionals.

**Table 5. Proportion of women experiencing mental health symptoms and recent IPV who used services in the 12 months prior to 10-year follow-up (n = 1507).**

|  | Counselling Service | Telephone help line | Financial counselling | Legal service |
|---|---|---|---|---|
|  | % (95% CI) | % (95% CI) | % (95% CI) | % (95% CI) |
| Depression, anxiety and/or posttraumatic stress symptoms | 43.3 [36.3–50.3] | 13.3 [7.9–18.8] | 10.9 [6.1–15.7] | 21.2 [14.4–28.0] |
| Intimate partner violence | 42.2 [34.3–50.1] | 12.2 [5.8–18.6] | 14.4 [8.1–20.7] | 28.2 [19.6–36.8] |
| Mental health symptoms and intimate partner violence | 50.7 [39.8–61.5] | 17.5 [7.7–27.3] | 17.7 [7.9–27.4] | 33.3 [21.1–45.5] |
| Whole cohort | 21.7 [1.5–18.7] | 5.8 [0.9–4.0] | 4.9 [0.8–3.3] | 10.7 [1.2–8.3] |

The most commonly used advocacy services used by women experiencing mental health problems and/or IPV were counselling services (used by two in five women experiencing mental health problems and/or IPV) and legal services (used by almost one in four women experiencing IPV and one in five women experiencing mental health problems, Table 5). Telephone help lines and financial counselling services were used by relatively few women.

## Discussion

While many qualitative studies document barriers to disclosure of mental health issues and IPV, there are relatively few population-based studies documenting the extent to which women experiencing mental health problems and/or IPV seek assistance from services. Our study is the first to examine use of primary care, mental health and advocacy services in a sample of women with known exposure to IPV and robust measures of mental health disorder.

One in four women in the study reported depressive, anxiety or posttraumatic stress symptoms at ten years postpartum, and almost one in five had experienced IPV in the previous 12 months. The majority of women experiencing mental health problems and/or IPV had seen a general practitioner three or more times in the previous 12 months. Despite frequent contact with general practitioner services, women often chose not to discuss mental health problems or IPV with health professionals. Less than half of women scoring at or above the cut-off score for clinically significant disorder on the CES-D, BAI and PCL-C had discussed their mental health with a general practitioner in the previous 12 months. Even fewer women experiencing IPV reported that they had discussed violence in their home and/or relationship difficulties with a general practitioner. The latter results are in accord with the Australian Personal Safety Survey which found that only around a third of women experiencing recent IPV had discussed this with a general practitioner. In contrast, a New Zealand household survey found that around 13% of women experiencing recent (past year) physical and/or sexual IPV had spoken to a general practitioner about it [17, 35]. Data from the 2007 Australian National Survey of Mental Health and Wellbeing indicate that 40.7% of women (aged 16 to 85 years) with mental health disorders used health services for mental health problems in the 12 months prior to the survey, with higher use of health services by women aged 35 to 64 years. Rates of health service use in the general population were higher for those with affective disorders than for those with anxiety disorders (results were not disaggregated by gender) [36]. Surveys conducted in the United States and New Zealand suggest comparable rates of consultation with health professionals for people experiencing mental health disorders [37, 38], although caution must be exercised in making comparisons across different jurisdictions, health systems and survey instruments [36]. We were unable to identify comparable data from population-based studies examining health care utilization of mothers with known symptom profiles for common maternal mental health disorders or known exposure to IPV.

As noted above, many factors may influence uptake of health services. In the Australian health care system, general practitioners are the first point of contact with health care.

Although visits to general practitioners are subsidised by Medicare (Australia's universal health insurance scheme), many practices charge above the schedule fee resulting in variable 'out of pocket costs' at the point of service. People who hold a Health Care Card generally do not incur 'out of pocket costs' for general practitioner services. Mental health care is primarily provided by psychologists working in private practice. Access to government subsided visits with a psychologist is facilitated via general practitioner referral. However, there are often long waiting lists and 'out of pocket' costs for these services. It is not surprising that a higher proportion of women had talked to a general practitioner about their mental health, than had talked to a mental health professional. It is likely that some women experiencing mental health problems may have been offered referral to a psychologist, but not been able to afford co-payments, or declined for other reasons. Although more women had seen a general practitioner than had visited a psychologist or other mental health professional in the previous 12 months, a higher proportion of women experiencing IPV had discussed relationship difficulties or IPV with a mental health professional than with a general practitioner. This may reflect the longer consultation format of visits with a mental health professional or it may be that the focus on mental health provides a context in which disclosure of relationship difficulties and/or IPV is more likely. Counter to our hypothesis that women experiencing physical IPV would be more likely to seek assistance from health professionals than women experiencing emotional IPV alone, women experiencing physical IPV were not more likely to visit general practitioners, and no more likely to discuss IPV with either a general practitioner or mental health professional compared with women experiencing emotional IPV alone suggesting that other factors determine women's uptake of health care.

## Strengths and limitations

Strengths of this study include: a community-based sample, standardised measures of maternal depressive, anxiety and posttraumatic stress symptoms and IPV, and collection of a range of information regarding contacts with health professionals and advocacy services. The IPV measure used was developed and validated in an Australian sample and has been widely used in clinical and population samples [39–41]. It provides a robust measure of exposure to physical and emotional IPV in the previous 12 months. Study limitations include: selection bias in the original sample and selective attrition over time. Both are likely to have biased prevalence estimates for mental health disorder and IPV downwards, the former because women experiencing mental health problems and/or IPV during pregnancy may be less likely to participate, and the latter because women experiencing mental health problems and/or IPV in the first year after birth were more likely to report mental health symptoms at 10 years postpartum [6]. While data are drawn from a prospective cohort, the analyses presented are cross-sectional and causal relationships cannot be established. Assessment of maternal mental health used robust, psychometrically validated tools, but did not involve standardised diagnostic interviews. It is likely that some women classified as experiencing clinically significant symptoms may not have met criteria for mental health disorder in a psychiatric assessment. It is equally likely that some women who scored below clinical cut-off scores may have been misclassified. It is unlikely that results would have changed significantly as a result of misclassification bias, but this factor does need to be considered when making comparisons between studies. The measure we used to assess post-traumatic stress symptoms (PCL-C) does not ask participants to report on types of trauma. It is possible, indeed likely, that symptoms may relate to a range of childhood and adult experiences of trauma including childhood maltreatment, IPV and birth trauma. Our findings are likely to generalise to other high-income countries with similar provision of universal primary health care and mental health care services, such as the UK and

Canada, but may not generalise to countries where access to primary health care and mental health care is limited to those with private health insurance. Data were not collected on presentations to hospital emergency departments, so we are unable to report on use of these services for mental health concerns and/or experiences of IPV. Finally, follow-up of the cohort at 10-years postpartum occurred prior to the onset of the COVID-19 pandemic. The social and economic consequences of the pandemic and pressures on health services are likely to have exacerbated the challenges for women seeking support for mental health problems and/or IPV [42, 43].

## Conclusions

General practitioners and mental health professionals play a key role in providing avenues for support and assistance to women experiencing mental health problems and IPV. In the tenth year after the birth of their first child, only a small minority of Australian mothers had not had contact with a general practitioner in the previous 12 months. While a majority of women experiencing depressive, anxiety or posttraumatic stress symptoms consistent with major disorder had talked to a general practitioner and/or a mental health professional about their mental health, one in three had not. The majority of women experiencing IPV had *not* disclosed this to a general practitioner or mental health professional.

Our findings underline the extent to which many women deal with mental health problems and/or IPV without the support that primary health care and mental health care services could provide. The extent of co-occurring IPV and mental health problems underlines the importance of more concerted and co-ordinated efforts to strengthen health systems responses. It is almost two decades since the WHO first called for a stronger response to IPV from the health sector [44] Our findings suggest that even before the COVID-19 pandemic, there was still much work to be done. As global attention is focused on the necessity and opportunities for systems change to address impacts of the pandemic on mental health and wellbeing [42], it will be important for policy makers and health services to reflect on what is already known about gaps in service accessibility and uptake to ensure practitioners are both supported and ready to do this challenging work [45, 46]. This requires leadership prioritising family violence as a health issue, greater recognition of IPV as a factor affecting women's mental health and tailoring of mental health care to social context and circumstances in which women find themselves. There is now good evidence that an advocacy approach; commitment to collaborative work practices; and ensuring that training, protocols and workplace support for health professionals undertaking this work is helpful [46].

## Supporting information

**S1 Table. Comparison of the original cohort and women who completed ten-year follow-up.**
(DOCX)

## Acknowledgments

We are extremely grateful to all of the women taking part in the study; to members of the Maternal Health Study Collaborative Group who contributed to the design of study instruments and data collection procedures for ten-year follow-up of mothers and children in the cohort; and to members of the Maternal Health Study research team who have contributed to data collection.

## Author Contributions

**Conceptualization:** Stephanie Brown.

**Data curation:** Deirdre Gartland.

**Formal analysis:** Deirdre Gartland.

**Funding acquisition:** Deirdre Gartland, Kelsey Hegarty, Stephanie Brown.

**Investigation:** Deirdre Gartland, Sandra Papadopoullos, Stephanie Brown.

**Methodology:** Deirdre Gartland, Kelsey Hegarty, Stephanie Brown.

**Project administration:** Deirdre Gartland, Sandra Papadopoullos, Stephanie Brown.

**Resources:** Deirdre Gartland, Kelsey Hegarty, Sandra Papadopoullos, Stephanie Brown.

**Software:** Deirdre Gartland.

**Supervision:** Stephanie Brown.

**Validation:** Deirdre Gartland, Stephanie Brown.

**Visualization:** Deirdre Gartland, Stephanie Brown.

**Writing – original draft:** Stephanie Brown.

**Writing – review & editing:** Deirdre Gartland, Kelsey Hegarty, Sandra Papadopoullos, Stephanie Brown.

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
