## [Decision Letter · Decision Letter 0]

16 Mar 2022

PONE-D-21-33064Patterns of health service utilisation of mothers experiencing mental health problems and intimate partner violence: ten-year follow-up of an Australian prospective mother and child cohortPLOS ONE

Dear Dr. Brown,

Thank you for submitting your manuscript to PLOS ONE. After careful consideration, we feel that it has merit but does not fully meet PLOS ONE’s publication criteria as it currently stands. Therefore, we invite you to submit a revised version of the manuscript that addresses the points raised during the review process.

Please consider carefully the reviewer´s comments about the manuscript.

We look forward to receiving your revised manuscript.

Kind regards,

José J. López-Goñi

Academic Editor

PLOS ONE

Journal Requirements:

(The funders had no role in study design, data collection and analysis, decision to publish, or preparation of the manuscript)

Please include your amended Funding Statement within your cover letter. We will change the online submission form on your behalf."

The Maternal Health Study has been supported by three project grants from the Australian National Health and Medical Research Council (NHMRC, 191222 awarded to SB; 433006 and 1048829 awarded to SB, DG and KH); grants from the William Buckland Foundation, ANZ Trustees Medical Research Fund and Australian Rotary Health, awarded to SB; the Safer Families Centre of Research Excellence (NHMRC, 1116690) awarded to KH and SB and the Stronger Futures Centre of Research Excellence (NHMRC 1198270) awarded to SB and DG. Research conducted at the Murdoch Children’s Research Institute is supported by the Victorian Government Operational Infrastructure Program. 

Reviewers' comments:

Reviewer's Responses to Questions

**Comments to the Author**

1. Is the manuscript technically sound, and do the data support the conclusions?

Reviewer #1: Yes

2. Has the statistical analysis been performed appropriately and rigorously? 

Reviewer #1: I Don't Know

3. Have the authors made all data underlying the findings in their manuscript fully available?

Reviewer #1: No

4. Is the manuscript presented in an intelligible fashion and written in standard English?

Reviewer #1: Yes

5. Review Comments to the Author

Reviewer #1: Thankyou for the opportunity to review this interesting and well written paper.

I have a small number of suggestions and questions

It would be helpful for non-Australian audiences to have a description of health system and pr8mary care services in Australia and the options for mental health support. In the discussion it would also be important to note that here are long waiting times for the medicare funded and indeed any counselling services.

It is not clear if the EPDS was one of the measures used at 10 years or was it question 10 the self -harm question used?

Did you also ask mothers about any emergency room visits? If not, this may have been helpful and could be noted in limitations.

The posttraumatic stress scale is used as a measure of mental ill health, but is this trauma only related to IPV? Increasingly in birth cohort studies PTSD will be linked to birth trauma but presumably 10 years down the track this is less the case. It would be good to include some discussion about the type of trauma women may have experienced since the birth 10 years previously.

The sample that completed the 10 year follow up were less likely to have experienced mental health problems or IPV following birth but can you say anything about the relationship between women who experienced mental health symptoms and or IPV in the first year after birth and mental health ten years later.

6. PLOS authors have the option to publish the peer review history of their article (what does this mean?). If published, this will include your full peer review and any attached files.

Reviewer #1: No

---

## [Author Response · Author response to Decision Letter 0]

7 Apr 2022

Reviewer #1 Response

Reviewer #1: Thankyou for the opportunity to review this interesting and well written paper.

1. I have a small number of suggestions and questions

It would be helpful for non-Australian audiences to have a description of health system and primary care services in Australia and the options for mental health support. In the discussion it would also be important to note that here are long waiting times for the medicare funded and indeed any counselling services.

Thank you for this suggestion. We have edited the following section of the discussion accordingly. 

In the Australian health care system, general practitioners are the first point of contact with health care. Although visits to general practitioners are subsidised by Medicare (Australia’s universal health insurance scheme), many practices charge above the schedule fee resulting in variable ‘out of pocket costs’ at the point of service. People who hold a Health Care Card generally do not incur ‘out of pocket costs’ for general practitioner services. Mental health care is primarily provided by psychologists working in private practice. Access to government subsided visits with a psychologist is facilitated via general practitioner referral. However, there are often long waiting lists and ‘out of pocket’ costs for these services.

2. It is not clear if the EPDS was one of the measures used at 10 years or was it question 10 the self -harm question used?

The full measure was used. Only data from item 10 are reported in the paper. 

3. Did you also ask mothers about any emergency room visits? If not, this may have been helpful and could be noted in limitations.

Mothers were not asked about emergency room visits in this stage of the study. We have noted this in limitations as follows.

Data were not collected on presentations to hospital emergency departments, so we are unable to report on use of these services for mental health concerns and/or experiences of IPV.

4. The posttraumatic stress scale is used as a measure of mental ill health, but is this trauma only related to IPV? Increasingly in birth cohort studies PTSD will be linked to birth trauma but presumably 10 years down the track this is less the case. It would be good to include some discussion about the type of trauma women may have experienced since the birth 10 years previously.

 The measure we used to assess post-traumatic stress symptoms (PCL-C) is designed to identify symptoms and does not ask participants to report on types of trauma. It is possible, indeed likely, that symptoms may relate to a range of childhood and adult experiences of trauma including childhood maltreatment, IPV and birth trauma. We have noted this in the paper as follows:

The measure we used to assess post-traumatic stress symptoms (PCL-C) does not ask participants to report on types of trauma. It is possible, indeed likely, that symptoms may relate to a range of childhood and adult experiences of trauma including childhood maltreatment, IPV and birth trauma. 

The sample that completed the 10-year follow up were less likely to have experienced mental health problems or IPV following birth but can you say anything about the relationship between women who experienced mental health symptoms and or IPV in the first year after birth and mental health ten years later.

 Women who experienced depressive and anxiety symptoms and or IPV in the first year after birth are more likely to report mental health symptoms at 10 years postpartum. We have updated the manuscript (page 17) to include this information. 

Editor’s feedback 

 We have reviewed the guidelines and updated the manuscript and file names accordingly. 

(The funders had no role in study design, data collection and analysis, decision to publish, or preparation of the manuscript)

The Maternal Health Study has been supported by three project grants from the Australian National Health and Medical Research Council (NHMRC, 191222 awarded to SB; 433006 and 1048829 awarded to SB, DG and KH); grants from the William Buckland Foundation, ANZ Trustees Medical Research Fund and Australian Rotary Health, awarded to SB; the Safer Families Centre of Research Excellence (NHMRC, 1116690) awarded to KH and SB and the Stronger Futures Centre of Research Excellence (NHMRC 1198270) awarded to SB and DG. Research conducted at the Murdoch Children’s Research Institute is supported by the Victorian Government Operational Infrastructure Program. We have included the following information in our covering letter.

The Maternal Health Study has been supported by three project grants from the Australian National Health and Medical Research Council (NHMRC, 191222 awarded to SB; 433006 and 1048829 awarded to SB, DG and KH); grants from the William Buckland Foundation, ANZ Trustees Medical Research Fund and Australian Rotary Health, awarded to SB; the Safer Families Centre of Research Excellence (NHMRC, 1116690) awarded to KH and SB and the Stronger Futures Centre of Research Excellence (NHMRC 1198270) awarded to SB and DG. Research conducted at the Murdoch Children’s Research Institute is supported by the Victorian Government Operational Infrastructure Program. 

There was no additional external funding received for this study. 

The funders had no role in study design, data collection and analysis, decision to publish, or preparation of the manuscript

We have included the following information in our covering letter. 

The investigator team welcome inquiries about the data and proposals for collaboration.

The conditions of ethics approval preclude data access via a public repository. Data sharing is subject to approval by the investigator team. Applications will be considered in context of papers in progress, and compliance with conditions of ethics approval and consent. Interested researchers are invited to contact the Principal Investigator, Professor Stephanie Brown (stephanie.brown@mcri.edu.au). 

4 We note that you have included the phrase “data not shown” in your manuscript. Unfortunately, this does not meet our data sharing requirements. PLOS does not permit references to inaccessible data. We require that authors provide all relevant data within the paper, Supporting Information files, or in an acceptable, public repository. Please add a citation to support this phrase or upload the data that corresponds with these findings to a stable repository (such as Figshare or Dryad) and provide and URLs, DOIs, or accession numbers that may be used to access these data. Or, if the data are not a core part of the research being presented in your study, we ask that you remove the phrase that refers to these data.

We have amended the manuscript to incorporate this information as follows:

In further analyses of these data, women experiencing physical IPV appeared to be no more likely to discuss IPV or relationship difficulties with their general practitioner (16.3% versus 17.3%, OR – 0.9, 95% CI 0.4-2.5) or mental health professional (30.6% versus 27.2%, OR = 1.4, 95% CI 0.6-3.1), compared with women experiencing emotional IPV alone.

---

## [Editor Report · Decision Letter 1]

25 May 2022

Patterns of health service utilisation of mothers experiencing mental health problems and intimate partner violence: ten-year follow-up of an Australian prospective mother and child cohort

PONE-D-21-33064R1

Dear Dr. Brown,

We’re pleased to inform you that your manuscript has been judged scientifically suitable for publication and will be formally accepted for publication once it meets all outstanding technical requirements.

Kind regards,

José J. López-Goñi

Academic Editor

PLOS ONE
---

## [Editor Report · Acceptance letter]

6 Jun 2022

PONE-D-21-33064R1 

Patterns of health service utilisation of mothers experiencing mental health problems and intimate partner violence: ten-year follow-up of an Australian prospective mother and child cohort 

Dear Dr. Brown:

I'm pleased to inform you that your manuscript has been deemed suitable for publication in PLOS ONE. Congratulations! Your manuscript is now with our production department. 

Kind regards, 

on behalf of

Dr. José J. López-Goñi 

Academic Editor

PLOS ONE